# Role of miR-181c in Diet-induced obesity through regulation of lipid synthesis in liver

Kei Akiyoshi[1], Gretha J. Boersma[2¤], Miranda D. Johnson[2], Fernanda Carrizo Velasquez[1], Brittany Dunkerly-Eyring[3], Shannon O'Brien[2], Atsushi Yamaguchi[4], Charles Steenbergen[3], Kellie L. K. Tamashiro[2]*, Samarjit Das[1,3]*

1 Department of Anesthesiology and Critical Care Medicine, Baltimore, MD, United States of America,
2 Department of Psychiatry & Behavioral Sciences, Baltimore, MD, United States of America, 3 Department of Pathology, Johns Hopkins School of Medicine, Baltimore, MD, United States of America, 4 Department of Cardiovascular Surgery, Saitama Medical Center, Jichi Medical University, Saitama, Japan

¤ Current address: GGZ Drenthe Mental Health Institute, Assen, The Netherlands
* ktamashiro@jhmi.edu (KLKT); sdas11@jhmi.edu (SD)

**Data Availability Statement:** We will provide all the raw data upon request or once the paper will get published.

## Abstract

We recently identified a nuclear-encoded miRNA (miR-181c) in cardiomyocytes that can translocate into mitochondria to regulate mitochondrial gene mt-COX1 and influence obesity-induced cardiac dysfunction through the mitochondrial pathway. Because liver plays a pivotal role during obesity, we hypothesized that miR-181c might contribute to the pathophysiological complications associated with obesity. Therefore, we used miR-181c/d$^{-/-}$ mice to study the role of miR-181c in hepatocyte lipogenesis during diet-induced obesity. The mice were fed a high-fat (HF) diet for 26 weeks, during which indirect calorimetric measurements were made. Quantitative PCR (qPCR) was used to examine the expression of genes involved in lipid synthesis. We found that miR-181c/d$^{-/-}$ mice were not protected against all metabolic consequences of HF exposure. After 26 weeks, the miR-181c/d$^{-/-}$ mice had a significantly higher body fat percentage than did wild-type (WT) mice. Glucose tolerance tests showed hyperinsulinemia and hyperglycemia, indicative of insulin insensitivity in the miR-181c/d$^{-/-}$ mice. miR-181c/d$^{-/-}$ mice fed the HF diet had higher serum and liver triglyceride levels than did WT mice fed the same diet. qPCR data showed that several genes regulated by isocitrate dehydrogenase 1 (IDH1) were more upregulated in miR-181c/d$^{-/-}$ liver than in WT liver. Furthermore, miR-181c delivered *in vivo* via adeno-associated virus attenuated the lipogenesis by downregulating these same lipid synthesis genes in the liver. In hepatocytes, miR-181c regulates lipid biosynthesis by targeting IDH1. Taken together, the data indicate that overexpression of miR-181c can be beneficial for various lipid metabolism disorders.

## Introduction

Lipogenesis during obesity is a major pathological condition, and there are many efforts to mitigate its harmful consequences [1]. Recently, altering microRNA (miRs) expression levels has been postulated as one way to reduce lipid production in the liver [2]. However, more

**Funding:** This work was supported by grants from the Maryland Stem Cell Research Fund, (grant number Mscrfd-4313), the National Institute on Aging (grant number U54AG062333), and the National Center for Advancing Translational Sciences (grant number U18TR003780), all awarded to SD. This work was also supported by a grant from the National Heart, Lung, and Blood Institute (grant number 5R01HL039752), awarded to CS. Finally, this work was also supported by the National Institute of Diabetes and Digestive and Kidney Diseases (grant number NIDDK P30 DK079637) awarded to KLKT. The funders had no role in study design, data collection and analysis, decision to publish, or preparation of the manuscript.

**Competing interests:** The authors have declared that no competing interests exist.

studies are needed to delineate the relevant miRs involved in this process, and their mechanism of regulation.

We have previously studied one particular miR, miR-181c, which regulates mitochondrial gene expression and alters mitochondrial function [3–5]. Overexpression of miR-181c in rats causes oxidative stress, which leads to cardiac dysfunction [3], similar to the reported upregulation of miR-181c in human cardiac dysfunction [6]. Recently, we showed that myocardial lipid accumulation from a high-fat (HF) diet upregulates miR-181c expression in the heart [7]. Additionally, mitochondria activate reactive oxygen species (ROS) production in a type 2 diabetes (T2D) diet-induced obesity (DIO) model, indicating a significant role for mitochondrial dysfunction in glucose homeostasis and insulin resistance [8–10]. As mitochondrial function changes, multiple cellular and physiological functions are affected, contributing to the development of common T2D-related complications.

One miRNA can have different targets in different organs [11,12]. miR-181c is expressed mainly in the mitochondrial fraction of heart, and targets a mitochondrial gene (cytochrome c oxidase subunit I, mt-COX1) [3–5]. However, it is possible that miR-181c does not localize to the mitochondria in other organs. In fact, it has been shown that miR-181c targets genes in a gastric cancer cell line, suggesting that it mainly localizes to the cytosolic compartment in the gastrointestinal tract [13]. Similarly, miR-181c was found to interact with tumor necrosis factor α in hematopoietic progenitor cells, further suggesting non-mitochondrial localization [14]. It has been demonstrated that miR-181c plays an important role in the priming phase of liver regeneration [15] and that overexpression inhibits Hepatitis C virus replication by directly targeting the 3′-untranslated region (UTR) of homeobox A1 [16].

Normal homeostasis requires a balance between lipid synthesis and lipid oxidation to prevent lipid deposition. In obesity and T2D, lipids such as triglycerides and total cholesterol are elevated [17–20]. The role of miRNAs in lipid metabolism is well documented [21]. The liver-specific miRNA miR-122 has been shown to regulate lipid synthesis by targeting multiple mRNAs that are responsible for lipid biosynthesis, such as fatty acid synthase and acetyl-CoA carboxylases 1 and 2 [22–24]. Sterol regulatory element binding transcription factor 1 (SREBF1) regulates cholesterol homeostasis by downregulating ATP-binding cassette transporters [25]. miR-33 has been shown to alter cholesterol and HDL generation by targeting SREBF1 mRNA [26]. Mitochondrial function directly influences lipid biosynthesis [27–29].

The aim of this study was to identify the role of miR-181c in liver lipid metabolism in the context of obesity. We took advantage of our global miR-181c/d$^{-/-}$ (c/d KO) mouse model, which allowed us to investigate liver lipid metabolism in the absence of miR-181c during DIO. Based on previous finding that the hearts of c/d KO mice are protected against ischemia-reperfusion injury [5] and diet-induced obesity associated cardiac dysfunction [7], we hypothesized that c/d KO mice would also be protected from HF-induced metabolic stress in the liver.

## Material and methods

### Animals

Male wild-type (WT) C57BL/6J mice (Jackson Laboratories, Bar Harbor, ME) and previously described c/d KO mice [5] were used where indicated. Mice were provided *ad libitum* access to standard laboratory chow (CH; 2018 Teklad, Envigo, Frederick, MD) or purified HF diet (60% fat, D12492, Research Diets, New Brunswick, NJ) and tap water. The mice were divided into four experimental groups: WT-CH, WT-HF, c/d KO-CH, and c/d KO-HF (n = 6–7 per group). In a subset of mice on the HF diet, a glucose tolerance test was performed at 29 weeks of age. A separate cohort of WT mice was used for the miR-181c overexpression study as described below. All mice were housed in standard polycarbonate cages in a humidity- and

temperature-controlled vivarium on a 12 h:12 h light:dark cycle with light onset at 6 am. All procedures were approved by the Institutional Animal Care and Use Committee at the Johns Hopkins University School of Medicine.

## Indirect calorimetry

Energy expenditure, respiratory exchange ratio, locomotor activity, and food intake of WT and c/d KO mice were determined by taking indirect calorimetric measurements in an open-flow indirect calorimeter (Oxymax, Columbus Instruments) at 0, 10, and 20 weeks of HF exposure. Data were collected for 3 days to confirm acclimation to the calorimetry chambers (stable body weights and food intake). Data from the third day were analyzed. Rates of oxygen consumption ($VO_2$, ml·kg$^{-1}$·h$^{-1}$) and carbon dioxide production ($\dot{V}CO_2$, ml·kg$^{-1}$·h$^{-1}$) were measured for each chamber every 16 min throughout the study. The respiratory exchange ratio (RER = $VCO_2$/$VO_2$), calculated by Oxymax software (v. 4.02), was used to estimate relative oxidation of carbohydrate (RER = 1.0) versus fat (RER approaching 0.7), not accounting for protein oxidation.

## Intraperitoneal glucose tolerance test

Prior to the intraperitoneal glucose tolerance test (IPGTT) the mice were fasted overnight (food was removed at 6 pm). Mice were moved into the testing room 1.5 h before glucose injection for habituation (8 am). The IPGTT was performed according to methods previously described [30]. A baseline blood sample was taken (9:30 am) via a small nick of the tail. Then, the mice were injected intraperitoneally with 1.5 mg/g glucose (20% glucose in sterile water solution; 10 am). Additional blood samples (20 μl) were taken 15, 30, 45, 60, and 120 min after glucose injection. Glucose levels in the blood were determined immediately with a handheld glucose analyzer (Freestyle; TheraSense, Alameda, CA, USA). Blood was collected into heparinized capillary tubes and stored in microcentrifuge tubes on ice. After the blood was centrifuged at 4˚C, plasma was collected and stored at -80˚C. Plasma insulin concentration was determined with a commercially available ultrasensitive mouse insulin ELISA kit (Crystal Chem, Downers Grove, IL, USA).

## Tissue collection

Animals were deeply anesthetized with isoflurane, and blood was collected via cardiac puncture. The heart and liver were rapidly dissected, flash-frozen in liquid nitrogen, and stored at -80˚C for further analysis. Blood was centrifuged at 4˚C and plasma was collected and stored at -80˚C.

## Hormone and triglyceride analyses

Plasma leptin levels were determined with a commercially available mouse leptin ELISA kit (EMD Millipore, St. Charles, MO, USA). The intra-assay variation of the kit was 1.06–1.76%, and the inter-assay variation was 3.01–4.59%. Plasma and liver triglyceride levels were determined with a commercially available colorimetric assay kit (Cayman Chemicals, Ann Arbor, MI, USA). For liver triglyceride levels, samples were further diluted (1:5) and used for triglyceride analysis according to the manufacturer's instructions. Intra- and inter-assay variations of the kit were 1.34% and 3.17%, respectively.

## Mitochondrial isolation protocol

Mitochondria were freshly isolated from hearts and livers by differential centrifugation [4]. Briefly, after incubation in RNAlater, the tissues were dissected and placed in Buffer A (in

mM: 180 KCl, 2 EGTA, 5 MOPS, 0.2% BSA; pH: 7.25). The tissues were then digested with trypsin (0.0001 g/0.1 g tissue) in 0.7 ml of ice-cold Buffer B (in mM: 225 mannitol, 75 sucrose, 5 MOPS, 0.5 EGTA, 2 taurine; pH: 7.25) and finally homogenized in Buffer B containing protease inhibitor cocktail (Roche Applied Science, Indianapolis, IN). To further separate the heart mitochondria from other cellular components and tissue debris, a series of differential centrifugations was carried out in a Microfuge 22R centrifuge (Beckman Coulter, Fullerton, CA) at 4˚C. The crude pellet was then lysed with QIAzol (Qiagen, Valencia, CA).

## qRT-PCR

Total RNA was isolated from hearts or liver tissue with an miRNeasy kit (Qiagen) for the miRNA-enriched fraction or an RNEasy kit (Qiagen) for a larger RNA fraction. For the mitochondrial miRNA-enriched fraction, we used a modified protocol [4]. In both cases, an RNase-free DNase kit (Qiagen) was used to eliminate genomic DNA contamination. To characterize the integrity of the isolated RNA, we performed spectrophotometric evaluation with a microspectrophotometer (Nanodrop, Thermo Scientific, Wilmington, DE). Only RNA samples for which the absorbance at 260 nm (A260) was greater than 0.15 were used for further experiments. The ratio of the readings at 260 nm and 280 nm (A260/A280) was used to evaluate the purity of the isolated RNA. For further and more accurate purity and integrity estimation of the isolated RNA, we profiled the samples in the Bioanalyzer 2100 (Agilent Technologies, CA). Only RNA with A260/A280 ~2.00 and RNA integrity number > 8 was used for the experiments. RNA from heart and liver were reverse transcribed with the miScript Reverse Transcription Kit (Qiagen). HiFlex and HiSpec buffers were used for mRNA and miRNA, respectively. PCR for miRNA and mRNA was carried out with an miScript SYBR green PCR kit (Qiagen) using a QuantStudio 5 (Thermo Scientific). qRT-PCR was carried out with primers for miR-181c, SNORD61, and 5S rRNA (Qiagen). All reactions were repeated in triplicate. Primer sequences for all other genes are reported in Table 1.

## Western blot analysis

Liver, heart, and isolated mitochondrial pellets were lysed with RIPA buffer (Cat. #9806S, Cell signaling Technology), and protein content was measured by Bradford assay. Cell homogenate protein was separated by 1D gel electrophoresis and then transferred to a polyvinylidene fluoride membrane. The membrane was blocked with 5% nonfat dry milk for 2 h at room temperature before being incubated overnight at 4˚C with antibodies to isocitrate dehydrogenase 1 (IDH1; 1:1000), isocitrate dehydrogenase 2 (IDH2; 1:1000; Cell Signaling, Danvers, MA), or α-

**Table 1. Primers used in qPCR analysis for gene candidates of interest.**

| Gene | Primer sequence (5′ → 3′) | |
|---|---|---|
| Mouse | Forward | Reverse |
| 16S rRNA | ACC GCA AGG GAA AGA TGA AA | GCC ACA TAG ACG AGT TGA TTC |
| ACLY | ACC AGA AGG GAG TGA CCA TC | GAT GTT GTC CAG CAT TCC AC |
| β-Actin | GGC TGT ATT CCC CTC CAT CG | CCA GTT GGT AAC AAT GCC ATG T |
| FASN | TGA GAT CCC AGC ACT TCT TG | TGA CAT GAA CAT TGG AGC CT |
| GAPDH | AGG TCG GTG TGA ACG GAT TTG | TGT AGA CCA TGT AGT TGA GGT CA |
| SREBP1 | AGC AGG AGA ACC TGA CCC TA | TTT CAT GCC CTC CAT AGA CA |
| 5S rRNA | TCT CGT CTG ATC TCG GAA GC | AGC CTA CAG CAC CCG GTA TT |
| IDH1 | ATG CAA GGA GAT GAA ATG ACA CG | GCA TCA CGA TTC TCT ATG CCT AA |
| IDH2 | GAC AAG CAC TAT AAG ACT GAC | TCT GGT GTT CTC GGT AAT G |

tubulin (Abcam, Cambridge, MA) in Tris-buffered saline (pH 7.4) containing 1% TWEEN-20 (TBS-T) and 1% bovine serum albumin (BSA). Membranes were incubated with the appropriate horseradish peroxidase–conjugated IgG secondary antibody in TBS-T with 1% BSA for 1 h at room temperature. Immunoreactive protein was visualized with an enhanced chemiluminescence analysis kit (GE HealthCare, Piscataway, NJ). The signals emitted for chemiluminescence were detected with the iBright Imaging System (ThermoFisher Scientific).

### Body composition analysis

After the mice were sacrificed, their body composition was determined by whole-body nuclear magnetic resonance imaging (EchoMRI, Echo Medical Systems, Waco, TX).

### Oil-Red-O staining

Liver samples were immediately frozen after collection. Samples were embedded in O.C.T for sectioning, and tissue sections were mounted on glass slides. The mounted sections were stained with Oil-Red-O.

### miR-181c overexpression in WT mice

To overexpress miR-181c in WT mice, we packaged an miR-181c expression construct into an adeno-associated virus (serotype 8; AAV-8). Control scrambled miRNA ("Scr"; Cat# AA08-C-miR0001-MR14-100, GeneCopoeia Inc., Rockville, MD, USA) and overexpression miR-181c ("miR-181c OE"; Cat# AA08-MmiR3275-MR14-200, GeneCopoeia Inc.) were packaged into a hepatocyte-specific serotype, AAV-8 construct. C57BL/6J mice were fed a HF diet for 4 weeks before being injected with virus construct ($10^{11}$ viral particles in PBS; n = 8 mice per construct) and remained on the HF diet until sacrifice 6 weeks later. A total of 50 μl of scrambled or miR-181c oligonucleotide was injected retro-orbitally. Mice were tested in an IPGTT 5 weeks after virus injection and sacrificed 1 week later for collection of blood and tissues as described above.

### Statistical analysis

Data were analyzed by 1-way or 2-way analysis of variance (ANOVA), and Bonferroni or Tukey corrected *post hoc* tests were used where appropriate. All analyses were carried out in Prism 9 (GraphPad Software Inc., CA). A $p$ value $<0.05$ was considered statistically significant. Results are presented as mean ± standard error of the mean (mean ± SEM).

## Results

### Metabolic profile of mice lacking miR-181c/d

As shown previously, c/d KO mice were born in the expected Mendelian ratios and exhibited no body weight differences at birth compared to WT littermates [5]. Prior to HF exposure, c/d KO mice had higher oxygen consumption ($VO_2$; Fig 1A) than did WT mice ($p = 0.01$), but no differences in $CO_2$ production (Fig 1B). Thus, c/d KO mice had a lower RER, indicating a preferential mobilization of fat, rather than carbohydrates, for energy needs ($p<0.0001$; Fig 1C). In addition, c/d KO mice had increased energy expenditure compared to that of WT mice ($p = 0.01$; Fig 1D) on chow diet. However, after 10 and 20 weeks of HF access, no significant group differences remained in oxygen consumption, $CO_2$ production, RER, or energy expenditure between the WT and c/d KO mice (Fig 1A–1D). There were significant group x time interactions for $VO_2$ (F (2, 18) = 59.78, $p<0.0001$; Fig 1A), RER (F (2, 18) = 468.6, $p<0.0001$; Fig 1C), and energy expenditure (F (2, 18) = 78.54, $p<0.0001$; Fig 1D).

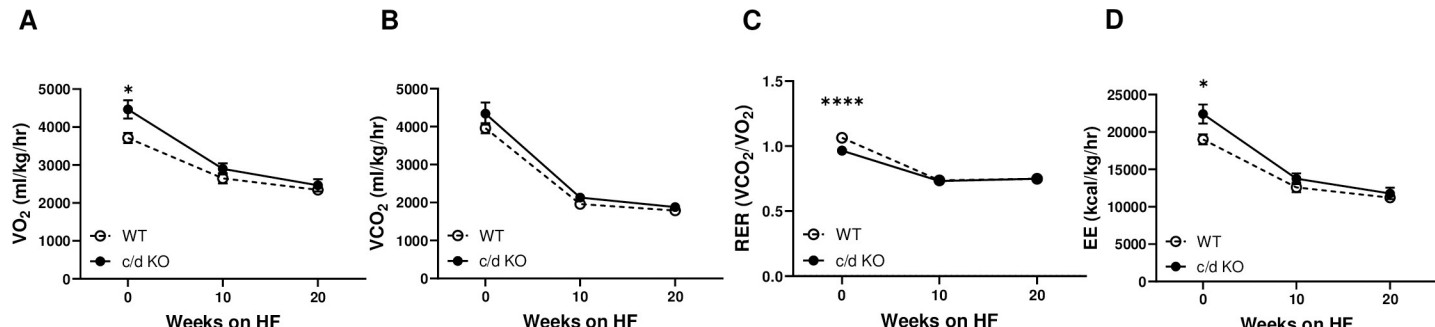

**Fig 1. Metabolic profiling of miR-181c/d$^{-/-}$ (c/d KO) mice.** (**A**) Oxygen consumption (VO$_2$), (**B**) carbon dioxide production (VCO$_2$), (**C**) respiratory exchange rate (RER), and (**D**) energy expenditure (EE) were measured before and during the 26 weeks of high-fat (HF) diet. Data are expressed as mean ± SEM (n = 4/genotype). $^*p < 0.05$ by Bonferroni *post hoc* analysis after intergroup differences were found by 2-way ANOVA. $^*p < 0.05$, $^{****}p < 0.0001$ by two-sample t-test.

## Lack of miR-181c leads to the accumulation of white adipose tissue and impaired glucose tolerance after HF exposure

While on chow diet (6–8 week of age), c/d KO and WT mice exhibited no differences in body weight (Fig 2A). When placed on the HF diet, both WT and c/d KO mice increased in body weight compared to their respective chow controls (F (3, 48) = 5.315, *p* = 0.003; Fig 2A). After 26 weeks of chow or HF exposure, there was a significant genotype diet effect on body fat (F (3, 37) = 30.83, *p*<0.0001; Fig 2B). Bonferroni *post hoc* analysis showed that c/d KO-HF mice had a higher body fat percentage than did WT-CH, c/d KO-CH, and WT-HF mice. In addition, there were significant effects of genotype and diet with respect to lean mass (F (3, 40) = 28.44, *p*<0.0001; Fig 2C). c/d KO-CH mice had lower lean body mass than did WT-CH mice, but not WT-HF mice. When exposed to a HF diet for 26 weeks, c/d KO-HF mice had lower lean mass as a percent of total carcass weight compared to all groups.

Under baseline conditions when all mice were 6–8 weeks old and on a normal chow diet, light cycle postprandial blood glucose levels did not differ between c/d KO (149.0±4.4 mg/dl) and WT mice (147.4±7.4 mg/dl). After 26 weeks of a HF diet, fasting glucose levels did not differ between WT and c/d KO mice (Fig 3A). However, fasting plasma insulin levels were significantly higher in the c/d KO mice than in the WT mice (*p* = 0.03; Fig 3B). When challenged with a glucose tolerance test, there was an overall time by group interaction effect on plasma glucose levels (F (5, 72) = 24.21, *p*<0.0001; Fig 3A). Bonferroni *post hoc* correction analysis

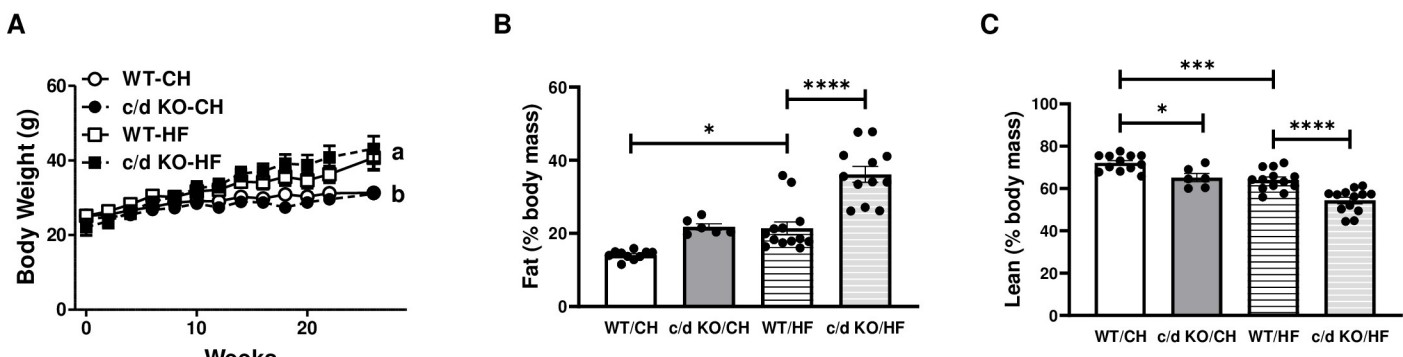

**Fig 2. Effect of high-fat (HF) diet on miR-181c/d$^{-/-}$ (c/d KO) mice.** (**A**) Body weight (n = 4/genotype/diet), (**B**) percent fat mass (n = 8-13/genotype/diet), and (**C**) percent lean mass (n = 8-13/genotype/diet) were measured before and during the 26 weeks of HF diet. Data are expressed as mean ± SEM. $^*p < 0.05$ by Bonferroni *post hoc* analysis after intergroup differences were found by 1-way ANOVA. $^*p < 0.05$, $^{***}p < 0.001$, $^{****}p < 0.0001$ by two-sample t-test.

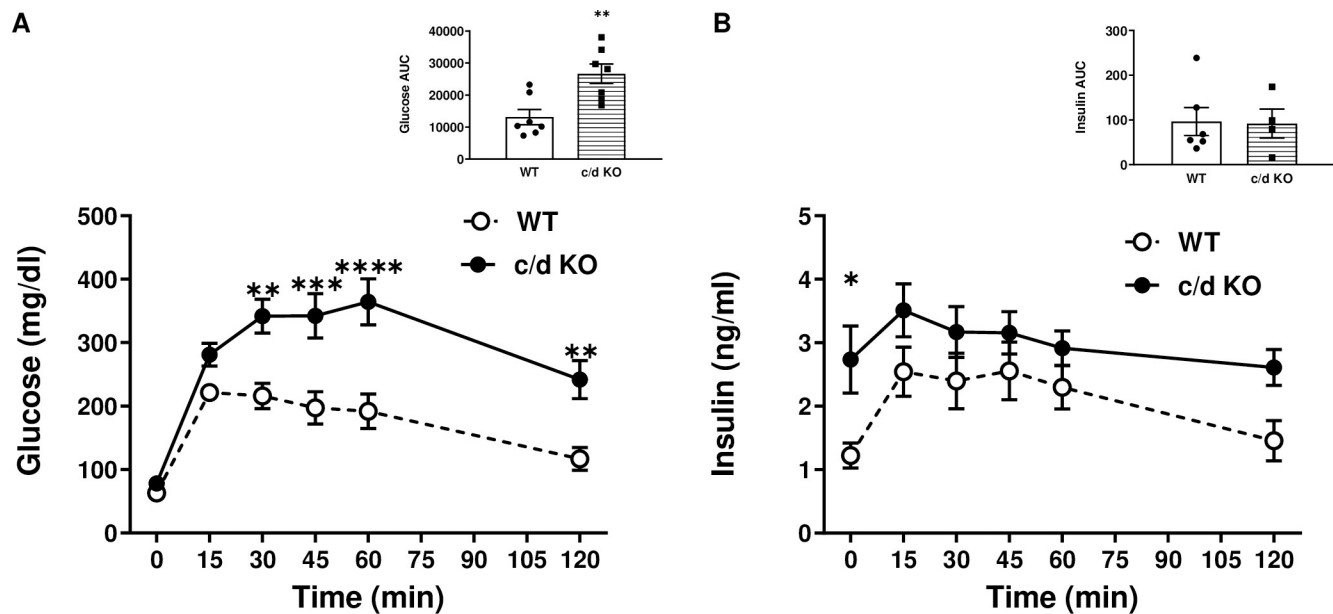

**Fig 3. Metabolic consequences of high-fat (HF) diet on miR-181c/d$^{-/-}$ (c/d KO) mice. (A)** Blood glucose and area under the curve during an intraperitoneal glucose tolerance test (IPGTT) and (**B**) plasma insulin levels and area under the curve during an IPGTT. Data are expressed as mean ± SEM and are from 6–7 mice/genotype/diet. *$p < 0.05$ by Bonferroni *post hoc* analysis after intergroup differences were found by 2-way ANOVA. *$p < 0.05$, **$p < 0.01$, ***$p < 0.001$, ****$p < 0.0001$ by two-sample t-test.

showed that at 30, 45, 60, and 120 min after glucose injection, circulating blood glucose levels were higher in c/d KO mice than in WT mice. In addition, c/d KO mice had an elevated area under the curve (AUC; $p = 0.004$) compared to that of WT mice. There was a main effect of group (F $(1, 71) = 18.47$, $p < 0.0001$) and time (F $(5, 71) = 2.714$, $p = 0.0266$) for plasma insulin levels (Fig 3B) but no statistically significant differences in the AUC between WT and c/d KO mice.

## Lack of miR-181c leads to impaired lipid biosynthesis in the liver after HF exposure

After 26 weeks of HF exposure, lipid accumulation in the liver of c/d KO mice was elevated compared to that in WT mice as measured by Oil-Red-O staining (Fig 4A). Plasma leptin level is considered a key biomarker for obesity and metabolic diseases [31,32]. Measurements showed no differences in plasma leptin between c/d KO and WT mice on normal chow diets (Fig 4B). However, after exposure to HF, both WT and c/d KO mice had elevated plasma leptin levels (F $(3, 19) = 53.43$, $p < 0.0001$). There was a main effect of diet (F $(3, 18) = 19.79$, $p < 0.0001$) on plasma triglyceride levels (Fig 4C). *Post hoc* analysis showed that plasma triglyceride levels of c/d KO-HF mice were elevated compared to those of WT-CH mice. *Post hoc* analysis also revealed that within the HF groups, liver triglyceride levels were significantly higher in the c/d KO than in the WT mice and respective chow-fed controls.

## Effect of HF diet on liver miR-181c expression

Recent data showed that a HF diet upregulates miR-181c in the heart [7]. To determine the regulation of miR-181c in the liver during diet-induced metabolic stress, we measured the expression of miR-181c in WT mice after exposure to 26 weeks of HF. Interestingly, miR-181c was decreased in the liver of WT mice after exposure to HF ($p = 0.02$; Fig 5A). In the heart,

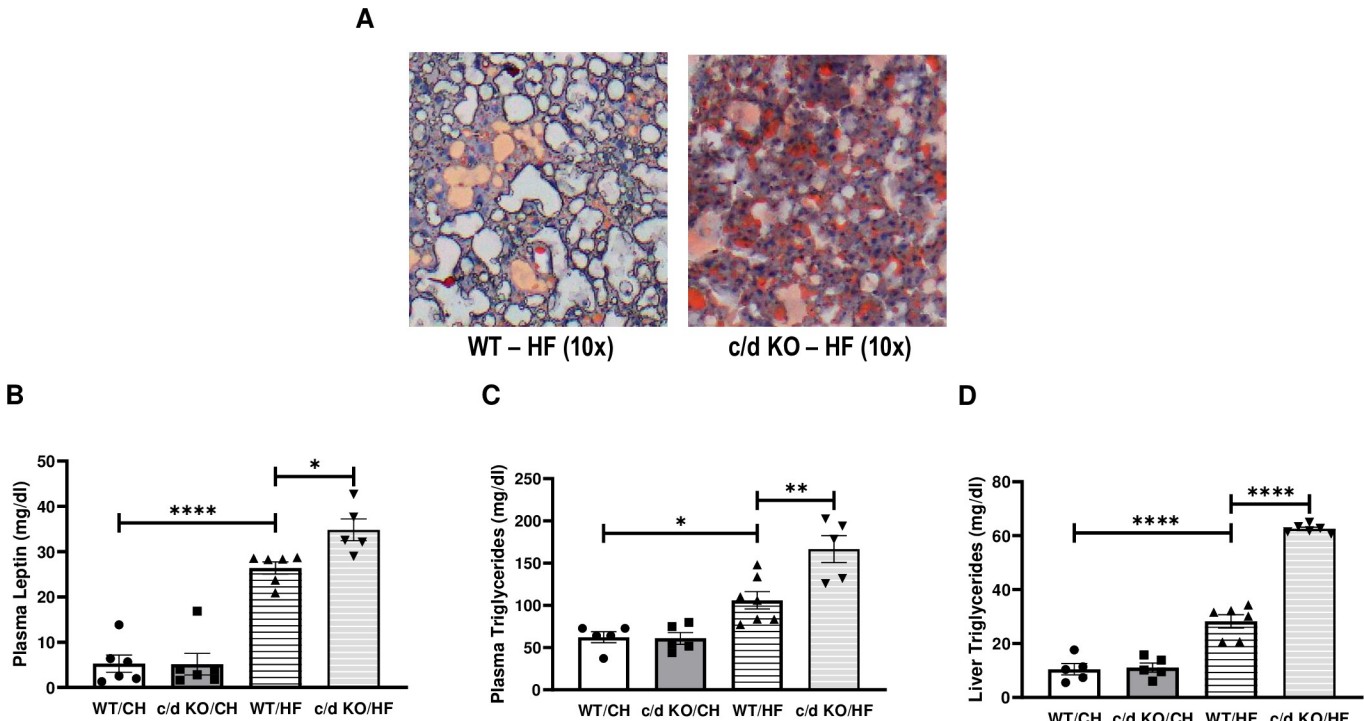

**Fig 4. Effect of high-fat (HF) diet on the liver of c/d KO mice.** (A) Liver Oil-Red-O staining for lipid droplets. Analysis of plasma leptin levels (B), plasma triglyceride levels (C), and liver triglyceride content (D) in WT and c/d KO groups fed normal chow or a HF diet. Data are expressed as mean ± SEM (n = 5–7 mice/genotype/diet). *$p < 0.05$ by Bonferroni *post hoc* analysis after intergroup differences were found by 1-way ANOVA. *$p < 0.05$, **$p < 0.01$, ****$p < 0.0001$ by two-sample t-test.

miR-181c mainly translocates to the mitochondrial fraction [3–5]. Whole tissue analysis in WT mice showed that expression of miR-181c was markedly higher in the liver than in heart tissue (Fig 5B). Next, to detect miR-181c in liver mitochondria, we isolated mitochondrial miRNA-enriched total RNA from both the heart and liver mitochondrial fraction. We detected significantly higher levels of miR-181c expression in the mouse heart mitochondria than in

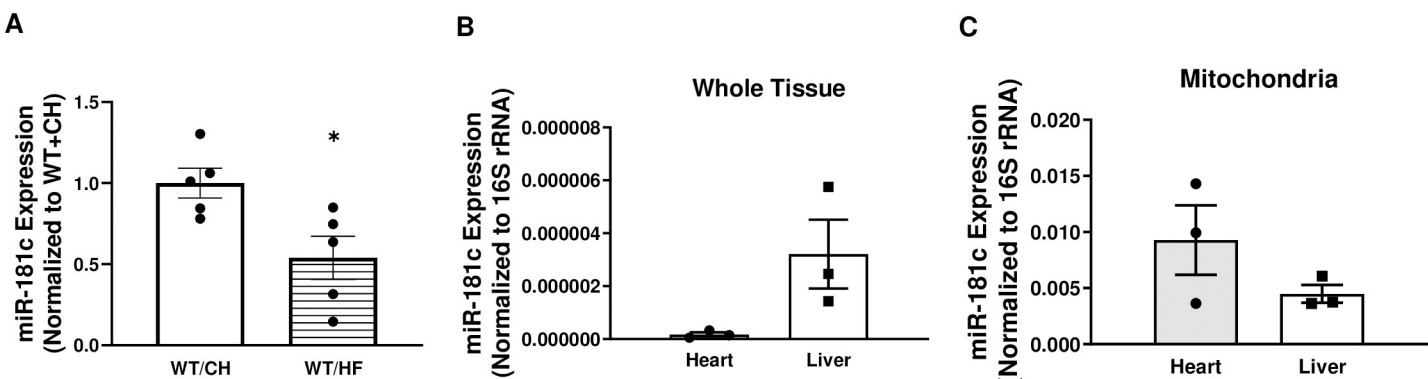

**Fig 5. Cytosolic localization of miR-181c in the liver.** Quantitative polymerase chain reaction (qRT-PCR) analysis of miR-181c expression in total RNA from the liver of normal chow (CH)-fed and high-fat (HF)-fed WT mice (A), in total RNA from the heart and liver tissue (B), and in the mitochondrial pellets of WT heart and liver tissue (C). Data are expressed as mean ± SEM (n = 3-7/genotype/diet). *$p < 0.05$ by t-test analysis after intergroup differences were found by 2-way ANOVA. *$p < 0.05$ by two-sample t-test.

liver mitochondria (Fig 5C). These data suggest that unlike heart, miR-181c is mainly expressed in the cytosolic fraction of liver.

## miR-181c targets IDH1 in the cytoplasm to inhibit lipid biosynthesis in the liver

Previously, it was shown that the "seed" sequence for miR-181 can directly bind to the 3′-UTR of the IDH1 gene at one putative site (UGAAUGU) [2]. In addition, overexpression of cytosolic-specific IDH (IDHc) resulted in elevated liver triglycerides and adipose tissue accumulation [33]. Because miR-181c is expressed mainly in the cytosolic fraction of the liver (Fig 5B and 5C) and IDHc is involved in lipid biosynthesis in the liver, we measured IDH protein expression in the liver of chow-fed WT and c/d KO mice. The expression of IDH1 is in the cytoplasm, whereas IDH2 is expressed in the mitochondrial fraction. Therefore, we compared the protein expression of both IDH1 and IDH2 in c/d KO mouse and WT liver. Loss of miR-181c led to the upregulation of IDH1 protein in liver ($p = 0.015$ versus WT mice; Fig 6A). However, IDH2 protein expression did not differ between c/d KO and WT mice (Fig 6B). These data suggest that miR-181c targets IDH1 mRNA in the cytoplasmic fraction of hepatocytes.

Finally, to determine potential mechanisms by which the IDH1 pathway might contribute to impaired metabolic functioning in the liver of mice on a HF diet, we measured liver mRNA expression of genes involved in lipogenesis in chow-fed WT and c/d KO mice (Fig 6C). We found that expression of mRNA for genes involved in *de novo* lipid synthesis (ATP citrate lyase [ACLY]; $p = 0.03$, fatty acid synthase [FASN]; $p = 0.03$) and fatty acid synthesis (SREBP1; $p = 0.03$) were increased in the livers of c/d KO mice compared to those of WT controls.

## Liver-specific miR-181c overexpression

Next, to determine if overexpression of miR-181c can protect against the liver-specific metabolic consequences of HF exposure, we used an AAV-8 package system. First, the dose of viral particles, retro-orbital vein delivery, and optimal time points were optimized (Fig 7A). After 4 weeks of HF diet, the mice were randomly selected for either AAV-8 scramble or AAV-8-miR-

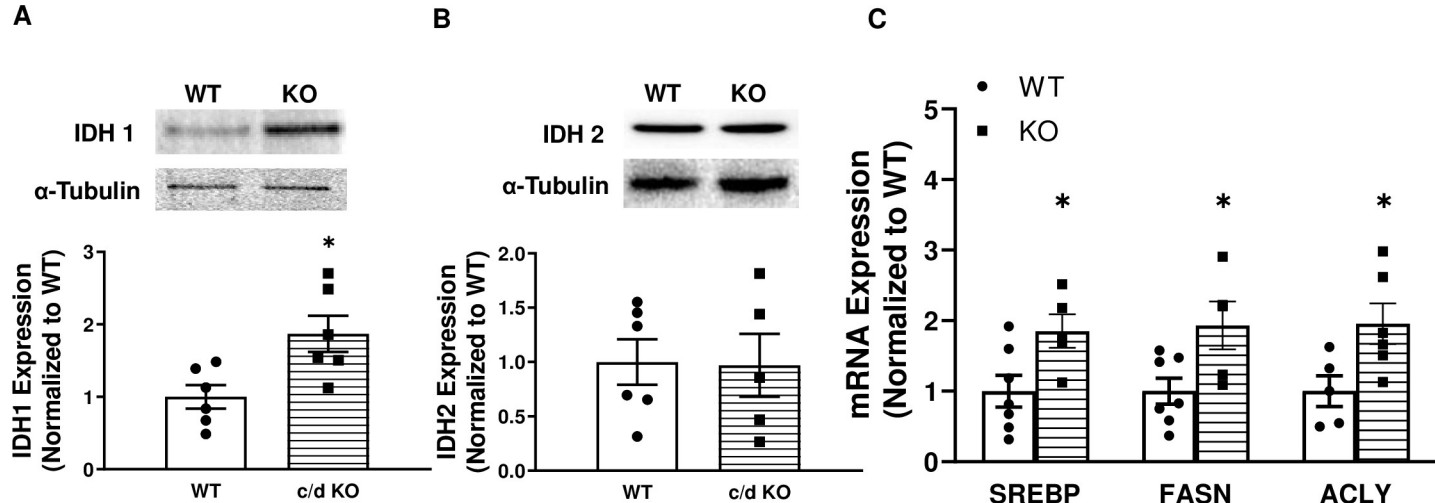

**Fig 6. In the absence of miR-181c, lipogenesis is activated by IDH1 upregulation.** Western blot analysis of protein lysate from liver IDH1 (**A**) and IDH2 (**B**). (**C**) Quantitative polymerase chain reaction (qRT-PCR) was used to assess liver mRNA expression of genes involved in lipogenesis. Sterol regulatory element binding transcription factor 1 (SREBP1), fatty acid synthase (FASN), and ATP citrate lyase (ACLY) were more highly expressed in WT than in c/d KO mice. Data are expressed as mean ± SEM (n = 5-7/genotype/diet). *$p<0.05$ by t-test analysis after intergroup differences were found by 2-way ANOVA. *$p<0.05$ by two-sample t-test.

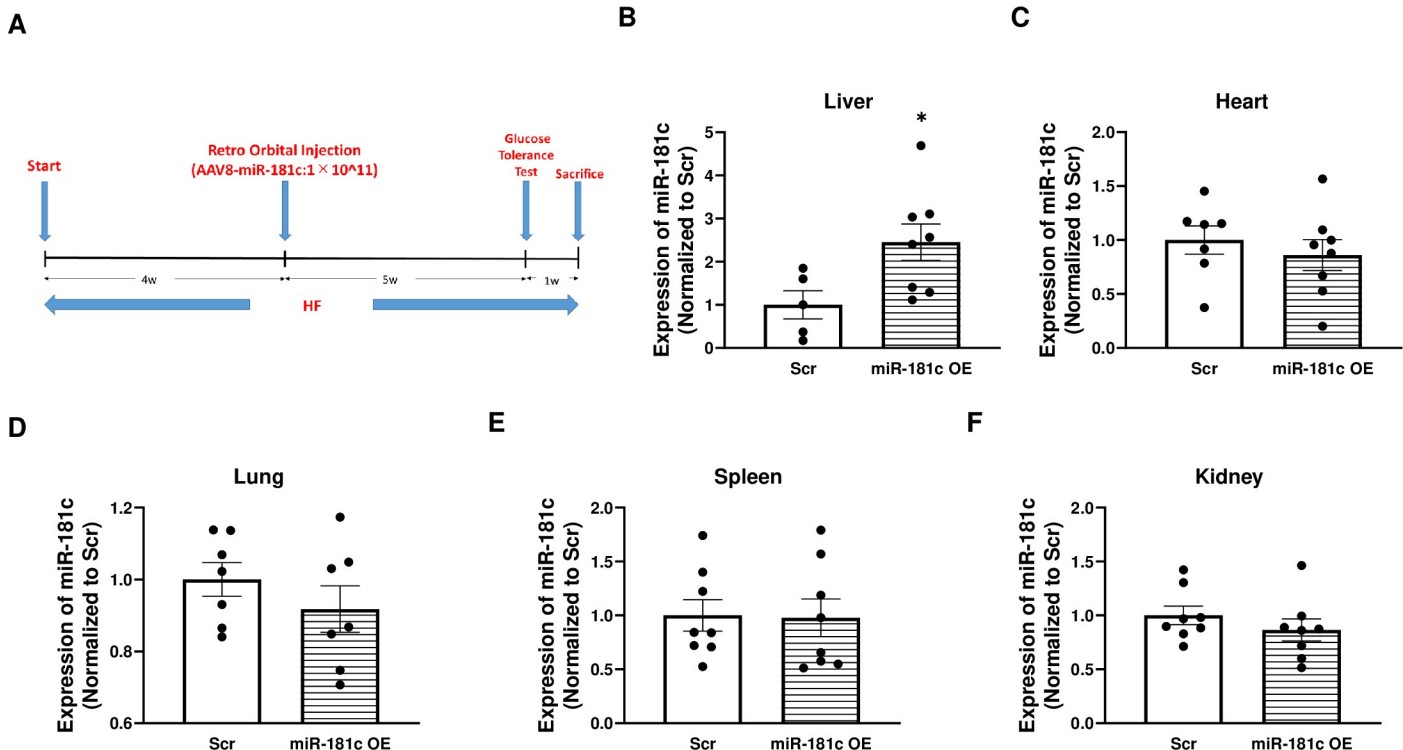

**Fig 7. Liver-specific miR-181c delivery and validation.** (A) *In vivo* miR-181c overexpression protocol. Quantitative polymerase chain reaction (qRT-PCR) analysis of miR-181c expression in total RNA isolated from liver (**B**), heart (**C**), lung (**D**), spleen (**E**), and kidney (**F**). OE, overexpression; Scr, scrambled oligonucleotide. Data are expressed as mean ± SEM; n = 8 for panels B-F. * $p<0.05$ by t-test analysis. * $p<0.05$ by two-sample t-test.

181c injection. Both groups of mice then maintained a HF diet for an additional 6 weeks. Using this treatment regimen of *in vivo* miR-181c delivery, we successfully overexpressed miR-181c in the liver by 2.5-fold ($p = 0.03$; Fig 7B). Importantly, miR-181c was not overexpressed in heart (Fig 7C), lung (Fig 7D), spleen (Fig 7E), or kidney (Fig 7F) after AAV-8-miR-181c injections.

Groups injected with AAV-8-miR-181c and AAV-8 scramble did not differ in body weight after 10 weeks of HF diet (Fig 8A). Similarly, after 10 weeks of the HF diet, there were no significant differences in baseline (0 min) blood glucose (Fig 8B) or insulin levels (Fig 8C) between groups injected with AAV-8-miR-181c and AAV-8 scramble. When the mice were challenged with a glucose tolerance test, we found an overall time by group interaction effect on plasma glucose level (Fig 8B). No circulating blood glucose or insulin level differences were present between AAV-8-miR-181c and AAV-8 scramble groups.

## Overexpression of miR-181c mitigates HF-induced lipogenesis

Liver-specific overexpression of miR-181c had no effect on IDH1 mRNA level (Fig 9A) but significantly downregulated IDH1 protein expression (Fig 9B). In contrast, miR-181c overexpression did not alter IDH2 expression at either the mRNA (Fig 9C) or protein level (Fig 9D) after 10 weeks of HF. Given our findings of the potential role for miR-181c in IDH1-regulated lipogenesis, we measured the expression of genes involved in lipogenesis in the liver. In mice that received 10 weeks of HF diet, treatment with miR-181c resulted in lower liver mRNA expression of genes involved in long chain fatty acid synthesis (FASN, $p = 0.03$; SREBP1, $p = 0.003$; Fig 9E).

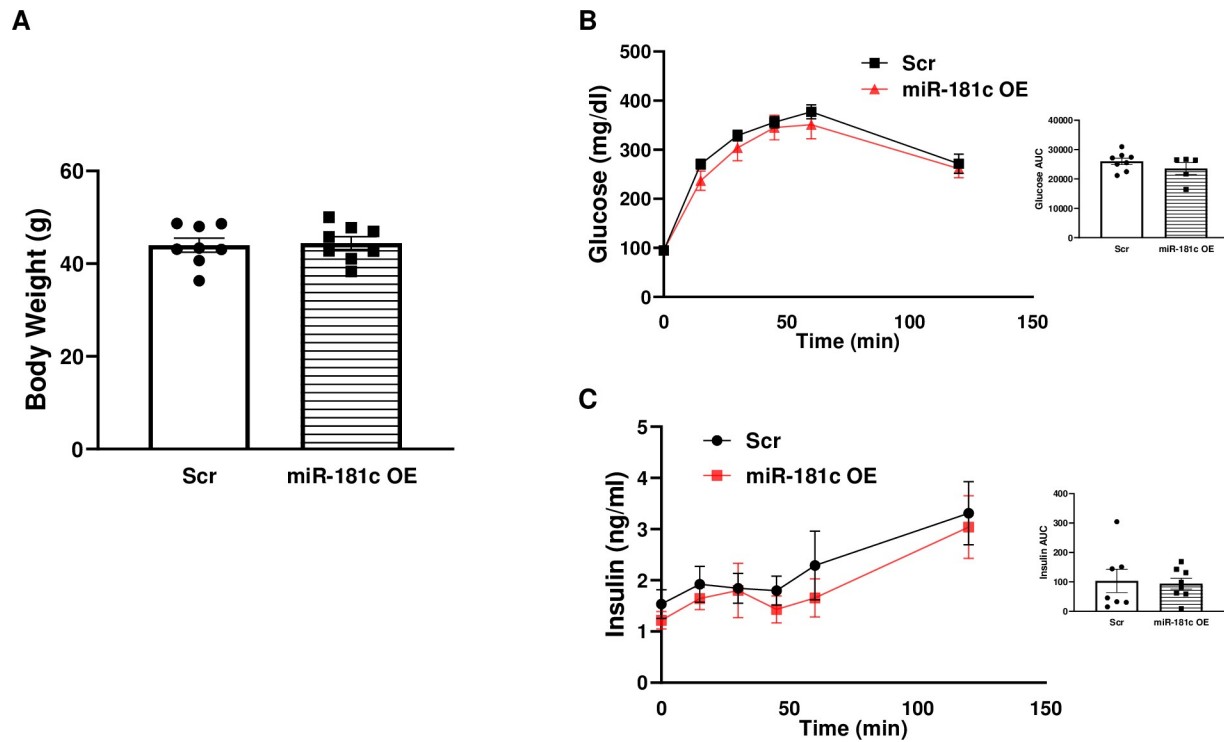

**Fig 8. Effect of miR-181c overexpression in liver on insulin secretion.** (**A**) Body weight. (**B**) Blood glucose and area under the curve during an IPGTT. (**C**) Plasma insulin levels and area under the curve during an IPGTT. OE, overexpression; Scr, scrambled oligonucleotide. Data are expressed as mean ± SEM; n = 8 per group. $^{*}p<0.05$ by Bonferroni *post hoc* analysis after intergroup differences were found by 2-way ANOVA.

The miR-181c-overexpressing mice and their AAV-8 scramble-injected controls exhibited no difference in liver weight after 10 weeks of a HF diet (Fig 10A). Histology showed elevated lipid accumulation in the liver of AAV-8 scramble-injected mice by Oil-Red-O staining, Masson-Trichrome, and H&E (Fig 10B upper panel) after 10 weeks of HF diet; however, AAV-8-miR-181c-injected mice had significantly lower lipid droplet accumulation (Fig 10B lower panel). No other changes were observed in either group by Masson-Trichrome or H&E staining (Fig 10B). *Post hoc* analysis revealed that within the HF groups, liver triglyceride levels were significantly lower in the miR-181c-overexpressing mice than in the controls (Fig 10C). Finally, among the HF group, liver-specific overexpression of miR-181c lowered the plasma triglyceride (Fig 10D) and plasma leptin (Fig 10E) levels compared to those of AAV-8 scramble injected mice.

## Discussion

The ability of miR-181c overexpression to confer robust protection in the obesity paradigm has two important implications. First, these results support miR-181c as a potential therapeutic target to combat obesity and lipogenesis. As it has already been established that miR-181c can be delivered *in vivo* using a nanovector [3], these findings identify a paradigm useful for testing the efficacy of miR-181c overexpression and determining its role in fat inhibition and cholesterol biosynthesis. Second, these findings demonstrate that compensation for the loss of miR-181c can confer protection against obesity when mice are exposed to a HF diet. c/d KO mice have a severe obese phenotype when fed a HF diet. Plasma leptin level is considered one of the

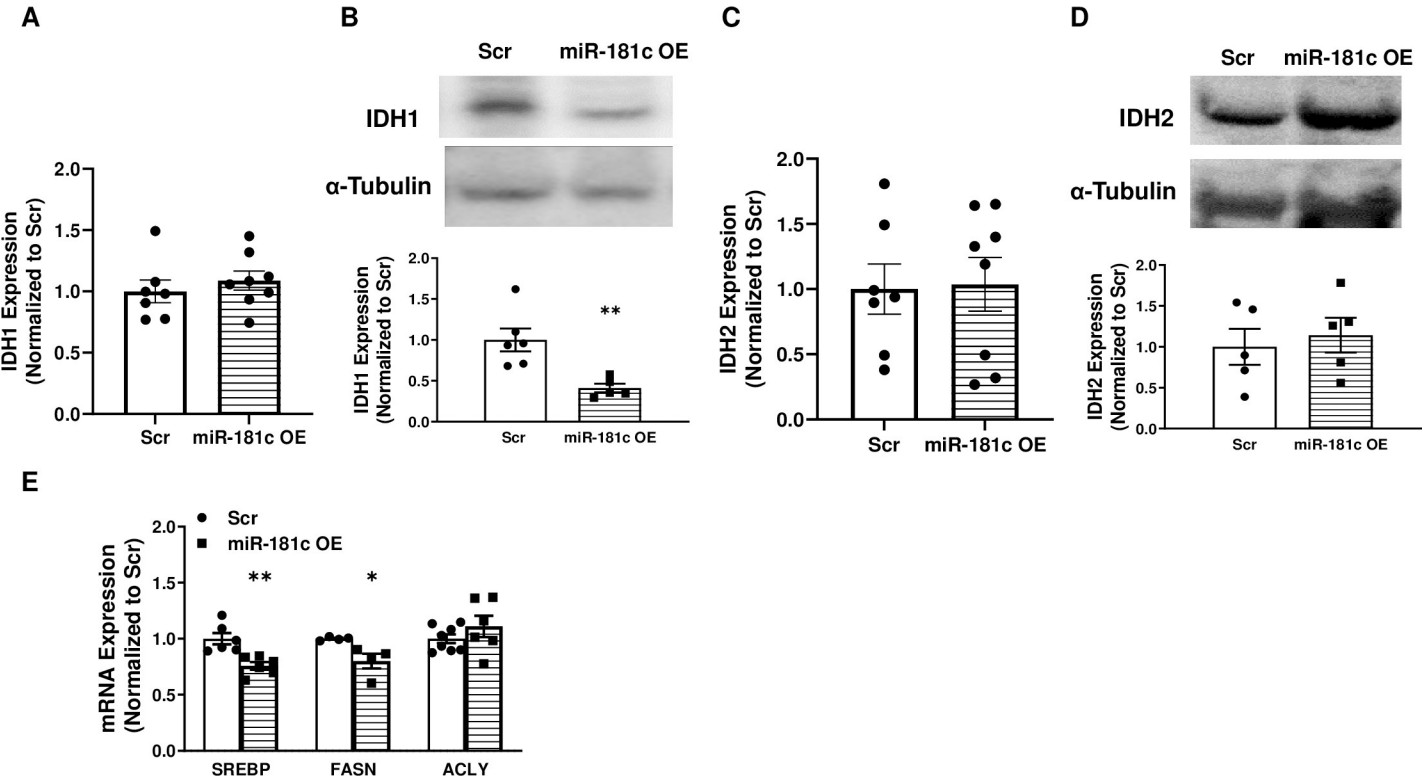

**Fig 9. Effect of liver-specific miR-181c overexpression on lipogenesis.** Western blot (**A**) and quantitative polymerase chain reaction (qRT-PCR, **B**) of IDH1 in the liver from AAV-8 vector-injected mice. Western blot (**C**) and qRT-PCR (**D**) of IDH2 in the liver of AAV-8 vector-injected mice. (**E**) qRT-PCR was used to assess liver mRNA expression of sterol regulatory element binding transcription factor 1 (SREBP1), fatty acid synthase (FASN), and ATP citrate lyase (ACLY) genes, which are involved in lipogenesis. OE, overexpression; Scr, scrambled oligonucleotide. Data are expressed as mean ± SEM; n = 8 per group. * $p < 0.05$ by t-test analysis. * $p < 0.05$, ** $p < 0.01$ by two-sample t-test.

key biomarkers for obesity and metabolic diseases [31,32]. We found significantly higher levels of plasma leptin in c/d KO mice than in WT mice after 26 weeks of HF diet (Fig 4B). However, liver-specific delivery of miR-181c significantly lowered the plasma leptin level compared to that of the scramble-injected group after 26 weeks of HF diet (Fig 10E). The data from the c/d KO mice and mice with liver-specific overexpression of miR-181c suggest that miR-181c can offer protection in the liver from HF exposure by inhibiting lipogenesis.

The nuclear-encoded miR-181 family members play an important role in cardiac function by regulating target genes both in the cytoplasm and within mitochondria. miR-181a/b regulates PTEN expression in the cytoplasm, whereas miR-181c regulates the mt-COX1 gene in mitochondria [5]. We have previously shown, both *in vitro* [4] and *in vivo* [3], a pivotal role for mitochondrial miR-181c in cardiac dysfunction. We [3–5] and others [6] have identified a significant role for miR-181c in the heart during end-stage heart failure. Unlike in heart, however, here we identified a nuclear-encoded cytoplasmic target, IDH1, for miR-181c in the liver (another mitochondria-enriched organ). Mature miR-181c can be expressed mainly in the mitochondrial fraction of cardiomyocytes; however, in hepatocytes, miR-181c directly binds to the 3′-UTR of IDH1 in the cytoplasm. By downregulating IDH1 during DIO, miR-181c can regulate lipid metabolism. These findings raise an important question: Why does miR-181c translocate into the mitochondria in the heart? Is it because cardiomyocytes lack IDH1 expression? Our current study can provide some important hints in the field of mitochondrial miRNA.

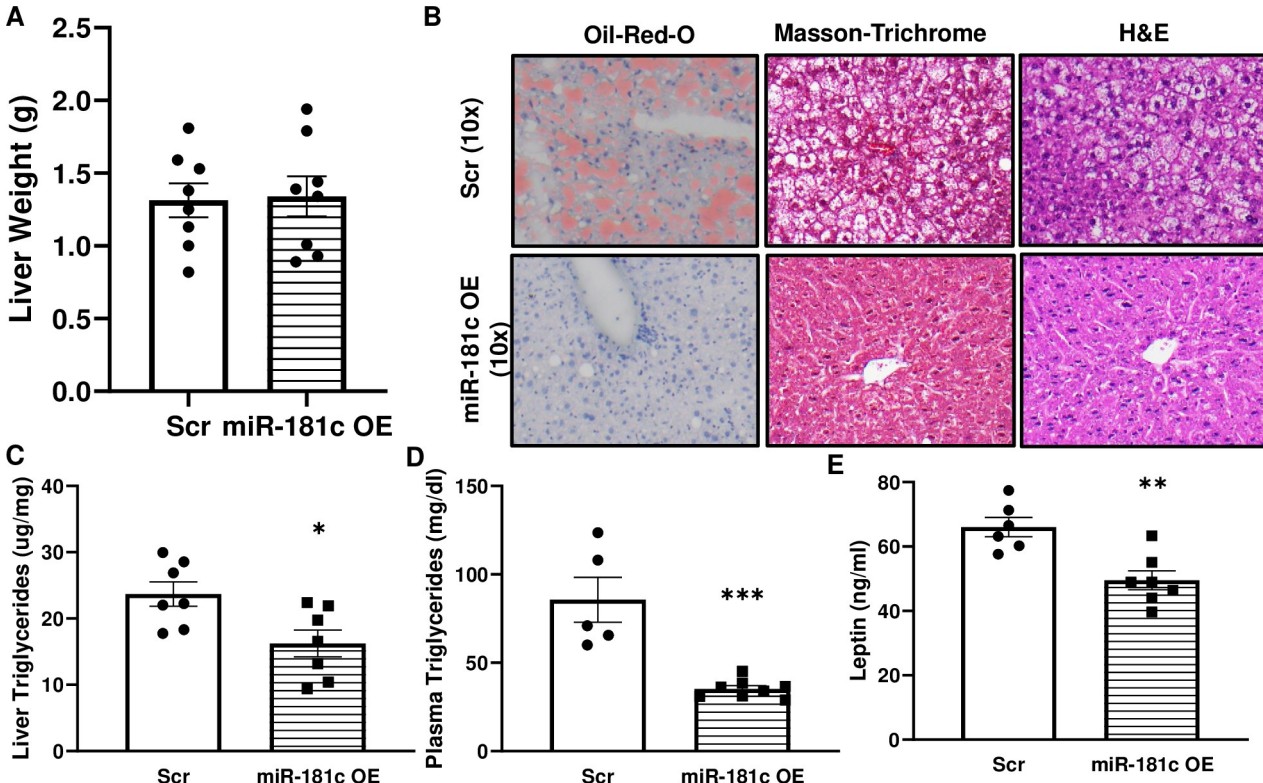

**Fig 10. Effect of liver-specific miR-181c overexpression on metabolic consequences of a high-fat (HF) diet.** AAV-8-miR-181c-injected and AAV-8-scramble-injected mice were compared after 10 weeks of a HF diet. **(A)** Total liver weight. **(B)** Liver histology, Oil-Red-O staining, Masson-Trichrome, and H&E staining. **(C)** Liver triglyceride content. **(D)** Plasma triglyceride levels. **(E)** Plasma leptin levels. OE, overexpression; Scr, scrambled oligonucleotide. Data are expressed as mean ± SEM; n = 8 per group. $^*p<0.05$ by Bonferroni *post hoc* analysis after intergroup differences were found by 1-way ANOVA. $^*p<0.05$, $^{**}p<0.01$, $^{***}p<0.001$ by two-sample t-test.

Acetyl-CoA and NADPH are essential common precursors and cofactors for lipid biosynthesis. IDH1, a key NADPH producer, contributes to the activation of triglyceride and cholesterol synthesis by the liver [33]. Similar to the metabolic phenotype of IDH1 transgenic mice [33], c/d KO mice have an increase in body fat composition and insulin insensitivity after 26 weeks of a HF diet. These two mouse models highlight the potential for IDH1 to promote induction of fatty liver, hyperlipidemia, and obesity by altering lipid biosynthesis in the liver, especially under HF-induced stress. Furthermore, IDH1 has been shown to activate lipogenesis under hypoxic conditions by reductive glutamine metabolism [34]. IDH1 can also regulate the lipogenic pathways by activating the transcriptional factors SREBP1 and SREBP2 [35]. Alternatively, Chu et al. [2] showed that knockdown of IDH1 downregulates lipid synthesis-related gene expression and upregulates β-oxidation and cholesterol transport-related gene expression, thereby inhibiting lipid synthesis in the liver. Therefore, liver-specific overexpression of miR-181c may be a potential therapeutic target for abnormal fat synthesis during DIO. miR-181c can attenuate lipid biosynthesis by directly binding to the 3′-UTR of IDH1 mRNA and perhaps overexpressing miR-181c in liver may play that therapeutic approach against obesity (Fig 11).

In this study, we observed that c/d KO mice exhibit a severe obese phenotype with significant hyperinsulinemia and hyperglycemia when they are fed a diet composed of 60% fat. These data also suggest that the miR-181c-IDH1 pathway plays an important role in dietary stress. Thus, liver-specific overexpression of miR-181c can protect against dietary stress by

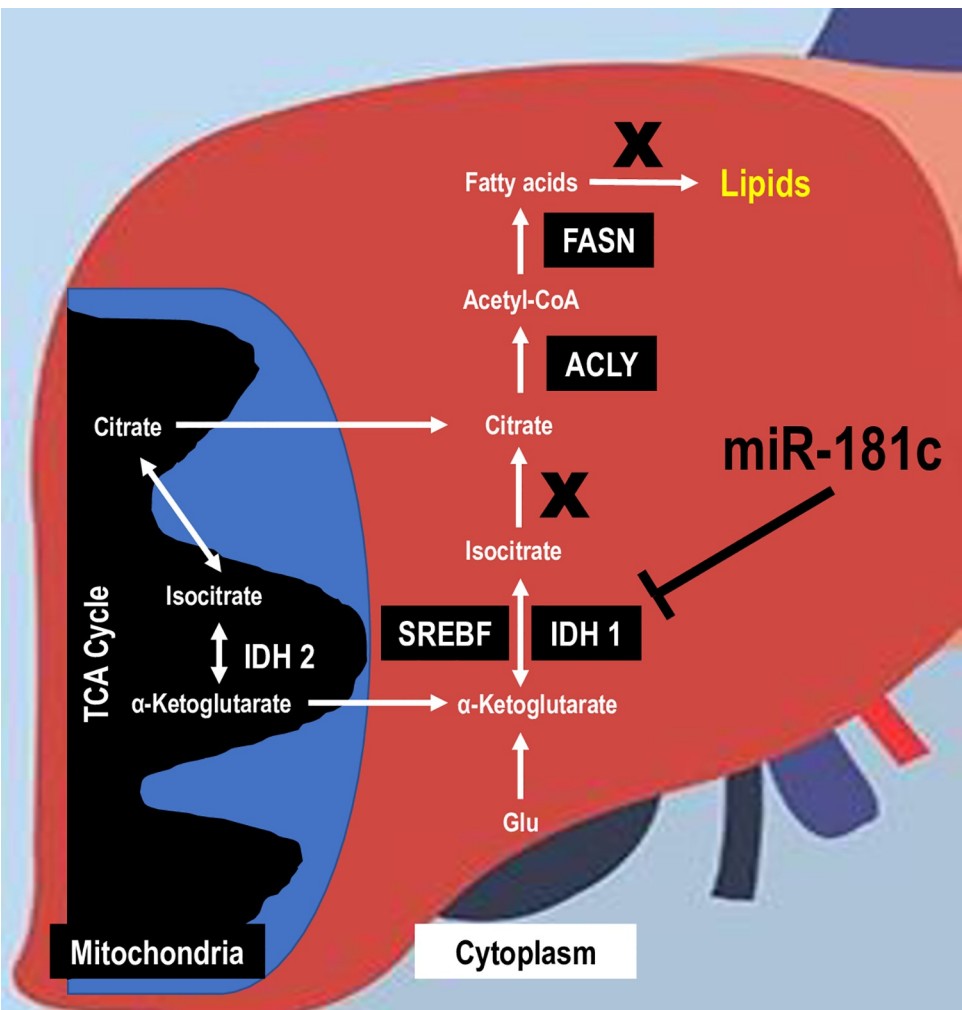

**Fig 11. Role of miR-181c in regulating high-fat diet-induced lipogenesis.** This schematic diagram illustrates key steps in the signaling pathway linking a high fat diet with increased lipogenesis. According to this model, high fat triggers lipogenesis by activating IDH1 though SREBF, ACLY, and FASN pathways. Liver-specific miR-181c can directly bind to the 3′-UTR of IDH 1 mRNA and mitigate lipogenesis.

inhibiting lipogenesis. miRNA therapeutics may be a potential treatment regimen; however, the tissue-targeted delivery is challenging with the potential for off-target side effects [36]. In the obesity field, miR-143~145 cluster delivery in the liver has been shown to have a protective effect against obesity-associated diabetes [37]. Previously, we showed that miR-181c can be overexpressed *in vivo* using an electrostatic complex: nanovector with positively charged liposomal nanoparticles and negatively charged plasmid DNA. Previous work also showed no immune response during this nanovector treatment regimen [3]. However, nanovector-based miR-181c delivery can distribute miR-181c in all the tissues, including heart. Overexpression of miR-181c can cause severe cardiac dysfunction via mitochondrial mechanisms [3–5]. In this study, we packaged miR-181c construct in AAV-8 and injected $10^{11}$ viral particles/mouse through retro-orbital injection, which did not cause miR-181c overexpressing in heart, lung, spleen, or kidney (Fig 7). Thus, future technologies should focus on liver-specific miR-181c delivery to prevent DIO complications, such as nonalcoholic fatty liver disease. Such interventions might be useful for patients with nonalcoholic fatty liver disease.

## Conclusions

In summary, in this study we demonstrated that miR-181c plays an essential role in inhibiting triglyceride and cholesterol biosynthesis by targeting IDH1 mRNA in the cytoplasm. c/d KO mice have a severe obese phenotype during HF. Overexpression of miR-181c during DIO can protect against the metabolic consequences of HF exposure by altering lipid metabolism.

## Supporting information

**S1 Fig.**
(PPTX)

**S2 Fig.**
(PPTX)

## Author Contributions

**Conceptualization:** Miranda D. Johnson, Fernanda Carrizo Velasquez, Brittany Dunkerly-Eyring, Charles Steenbergen, Kellie L. K. Tamashiro, Samarjit Das.

**Data curation:** Kei Akiyoshi, Gretha J. Boersma, Miranda D. Johnson, Fernanda Carrizo Velasquez, Brittany Dunkerly-Eyring, Shannon O'Brien.

**Formal analysis:** Kei Akiyoshi, Gretha J. Boersma, Miranda D. Johnson, Fernanda Carrizo Velasquez, Brittany Dunkerly-Eyring, Shannon O'Brien, Kellie L. K. Tamashiro.

**Funding acquisition:** Charles Steenbergen, Samarjit Das.

**Project administration:** Samarjit Das.

**Resources:** Samarjit Das.

**Supervision:** Atsushi Yamaguchi, Charles Steenbergen, Kellie L. K. Tamashiro, Samarjit Das.

**Validation:** Kei Akiyoshi.

**Writing – original draft:** Kei Akiyoshi, Miranda D. Johnson, Kellie L. K. Tamashiro, Samarjit Das.

**Writing – review & editing:** Gretha J. Boersma, Miranda D. Johnson, Fernanda Carrizo Velasquez, Brittany Dunkerly-Eyring, Shannon O'Brien, Atsushi Yamaguchi, Charles Steenbergen, Kellie L. K. Tamashiro, Samarjit Das.

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
