## [Decision Letter · Decision Letter 0]

8 Sep 2021

PONE-D-21-26341Role of miR-181c in Diet-Induced Obesity Through Regulation of Lipid Synthesis in LiverPLOS ONE

Dear Dr. Das,

Thank you for submitting your manuscript to PLOS ONE. After careful consideration, we feel that it has some merit but does not fully meet PLOS ONE’s publication criteria as it currently stands. Therefore, we invite you to submit a revised version of the manuscript that addresses the points raised during the review process.

Please carefully address all of the reviewers' concerns (see below).  Additional experiments or analysis should be performed to address criticisms from reviewer 1. Please submit your revised manuscript by December 6. If you will need more time than this to complete your revisions, please reply to this message or contact the journal office at plosone@plos.org. Please include the following items when submitting your revised manuscript:A rebuttal letter that responds to each point raised by the academic editor and reviewer(s). You should upload this letter as a separate file labeled 'Response to Reviewers'.A marked-up copy of your manuscript that highlights changes made to the original version. You should upload this as a separate file labeled 'Revised Manuscript with Track Changes'.An unmarked version of your revised paper without tracked changes. You should upload this as a separate file labeled 'Manuscript'.

We look forward to receiving your revised manuscript.

Kind regards,

Aijun Qiao, Ph.D.

Academic Editor

PLOS ONE

"This work was supported by grants from the MSCRF, Mscrfd-4313, U54AG062333 and U18TR003780 (S.D.), 5R01HL039752 (C.S.), JHU-UMD Diabetes Research and Training

Center (NIDDK P30 DK079637). "

Reviewers' comments:

Reviewer's Responses to Questions

**Comments to the Author**

1. Is the manuscript technically sound, and do the data support the conclusions?

Reviewer #1: Partly

Reviewer #2: Partly

2. Has the statistical analysis been performed appropriately and rigorously? 

Reviewer #1: Yes

Reviewer #2: Yes

3. Have the authors made all data underlying the findings in their manuscript fully available?

Reviewer #1: Yes

Reviewer #2: Yes

4. Is the manuscript presented in an intelligible fashion and written in standard English?

Reviewer #1: Yes

Reviewer #2: Yes

5. Review Comments to the Author

Reviewer #1: In this manuscript, Akiyoshi et al explored the role of miR-181c in diet induced obesity in vivo through both loss and gain of function assays. In miR-181c/d knockout mouse model, the authors have shown that inactivation of miR-181c/d led to glucose intolerance post 26 weeks high fat diet challenge associated with more lipid accumulation in liver and up-regulation of lipid synthesis genes potentially through up-regulation of IDH1 at protein level. Similarly, the authors have also tested the response of HFD in miR-181c overexpressing mouse model through AAV mediated gene delivery. Although no change in body weight, the authors claimed expression of miR-181c can attenuate lipid accumulation in liver post high fat diet challenge.

Previously the authors have shown inactivation of miR-181c/d protected mice against ischemia-reperfusion injury, therefore, the results presented in this manuscript are un-expected. Here are several concerns:

1. The regulatory role of miR-181c/d on IDH1

No sufficient data supporting that miR-181c directly regulates IDH1 in liver. For example, the authors have not shown mRNA expression level of IDH1 and IDH2 in miR-181c/d knockout animals or AAV8-miR-181c injected animals. There is no luciferase assay suggesting miR-181c indeed regulate IDH1 3’UTR activity. Without this evidence, the changes of IDH1 protein expression could be secondary effect.

2. Relevance of miR-181c in liver disease

The authors hypothesized that c/d KO mice would be protected from metabolic stress, so the results shown here are unexpected. Is there any change of miR-181c or miR-181d in NASH or other liver disease?

3. Characterization of mouse phenotype post high fat diet challenge

The authors should provide better phenotype characterization of the mouse model. For example, there is only one-time GTT and ITT. The authors should show baseline GTT and ITT in the mouse models (before HFD). There is no liver weight; H&E and liver fibrosis results.

Minor issue

1. The oil-red staining images have poor quality. No scale bar, Figure 4A is not white balanced.

Reviewer #2: The authors have explored the role of miR-181c in a DIO mouse model using a miR-181c -/- mouse and rescue with a AAV-8 delivering miR-181c.

Endpoints are:

Changes in target gene expression using qPCR

Indirect calorimetric measurements

Glucose tolerance tests/insulin levels

Body fat

TGs

Concluding that IDH1 is important in coordinating the changes in lipid biology related to 181c and propose that overexpression of 181c is protective

I do think the observations are of merit and warrant publication but I do have I number of reservations/comments:

- Be consistent in 181c nomenclature ie use 181c throughout

- The main text describing Figure 8, I personally found confusing in regards to glucose tolerance test and would just state no blood glucose or insulin levels were measured

- The authors measure leptin but no rationale as to why measured

- Why did the authors chose a retro-orbital route for AAV delivery, especially if aiming for liver expression, admittedly this was achieved, but tail vein injection is often used to optimize delivery to the liver.

- Could the authors explain why the numbers of mice in each group or observations, varied so much.

- Figure 4+10 (when any histology) need to include magnification

- Figure 5B +5C can you explain why in whole heart tissue you have low levels of 181c but then can find it in the mito fraction. Whole tissue still has the mito fraction within it, so would it not express 181c too?

- For first use of gene name in the manuscript define, as definition of gene names occurs later in the manuscript or in figure legends

- The evidence for IDH1 is mainly through association based on experimental observations and the literature. It would have been good if the authors had demonstrated that 181c is directly acting on the IDH1 3’UTR in a liver cell to provide more supporting evidence for the authors conclusion.

- Discussion, I would present some ideas about why 181c has a cytosolic v. mitochondrial location in the liver in contrast to the heart. Are any gender differences anticipated/known, as authors have used a male mouse model. I would be more reserved on the use of 181c as a protective agent for obesity as in the context of the liver it can act as an Oncomir too.

6. PLOS authors have the option to publish the peer review history of their article (what does this mean?). If published, this will include your full peer review and any attached files.

Reviewer #1: No

Reviewer #2: No

---

## [Author Response · Author response to Decision Letter 0]

1 Nov 2021

To, 

Dr. Aijun Qiao, Ph.D.

Academic Editor

PLOS ONE 

Dear Dr. Qiao: 

We thank the reviewers for their careful consideration, favorable comments, and many helpful suggestions for improving our manuscript. In addition, we would also like to thank you for giving us the opportunity to re-submit our manuscript to PLOS ONE. 

We have addressed all of the reviewers’ comments and their requests for additional data. Particularly, we have included qPCR results for IDH1 and IDH2 expression after miR-181c overexpression using the liver tissues; along with proper histopathological evaluation. Additionally, we have also included the liver weight after miR-181c overexpression using AAV-8 vector to respond to the concern raised by Reviewer #1. We have also included the gender difference in terms of miR-181c expression in the liver to respond to one of the points raised by Reviewer #2. We agree that gender differences are very important, and we are planning to further study the implications of higher miR-181c levels in female livers in our future studies. We hope that the reviewers agree that we have addressed all of their concerns, and trust this manuscript is now acceptable for publication in PLOS ONE. We look forward to hearing favorably from you. 

We have included detailed responses to each of the reviewer’s comments below. Their comments are in black and our responses are in red.

Reviewer #1:

1. The regulatory role of miR-181c/d on IDH1

No sufficient data supporting that miR-181c directly regulates IDH1 in liver. For example, the authors have not shown mRNA expression level of IDH1 and IDH2 in miR-181c/d knockout animals or AAV8-miR-181c injected animals. There is no luciferase assay suggesting miR-181c indeed regulate IDH1 3’UTR activity. Without this evidence, the changes of IDH1 protein expression could be secondary effect.

Thank you for the suggestion. In our knock-out mouse model, miR-181c is deleted from birth. Therefore, it is highly possible that a compensatory mechanism exists which directly or indirectly influences IDH1 expression in the liver, as evidenced by our previous studies 1,2. To validate whether miR-181c regulates IDH1 expression in the liver, we performed qPCR for IDH1 and IDH2 in the liver of AAV8-miR-181c injected mice. As shown in Fig. 9A-B, miR-181c overexpression significantly downregulates IDH1 protein expression, and there are no changes in IDH1 mRNA expression, suggesting that miR-181c translationally inhibits IDH1 expression by directly binding to the IDH1 3’-UTR region. We did not observe any changes in IDH2 expression (Fig. 9C-D). 

The reviewer is correct that the biochemical assay using dual luciferase would determine the direct association of a miRNA and its target mRNA. This assay was already performed in hepatocytes to determine whether the 3’-UTR of IDH1 is a direct target for miR-181a 3. The putative site (UGAAUGU) for miR-181a and miR-181c is identical, and that is the “seed” sequence for the miR-181 family 1. Chu et al., 3 has demonstrated that the miR-181 “seed” sequence can directly bind to the 3’-UTR of IDH1 mRNA. Even though the conclusion of the Chu et al. study3 is different than our current study, the dual-luciferase assay is identical to what the reviewer has suggested. 

2. Relevance of miR-181c in liver disease

The authors hypothesized that c/d KO mice would be protected from metabolic stress, so the results shown here are unexpected. Is there any change of miR-181c or miR-181d in NASH or other liver disease?

Thank you for the great suggestion. This would indeed suggest that miR-181c is a therapeutic candidate for liver diseases. In our study, we have provided strong evidence for the therapeutic role of miR-181c in obesity-induced lipogenesis. Furthermore, Mukherjee et al.4, has demonstrated the projective role of miR-181c in chronic liver disease. Similar to our study, Mukherjee et al., observed (in chronic liver disease due to Hepatitis C viral infection) significant downregulation of pre-miR-181c and miR-181c in the liver. In their study, the authors have proposed that miR-181c can be used as a therapeutic target for end-stage liver disease, including hepatocellular carcinoma (HCC), based on the protective role of miR-181c delivery to the liver 4. Non-Alcoholic SteatoHepatitis (NASH) is also considered an end-stage liver disease. It would be interesting to examine miR-181c regulation in NASH samples. However, NASH is beyond the scope of this current study. 

3. Characterization of mouse phenotype post high fat diet challenge

The authors should provide better phenotype characterization of the mouse model. For example, there is only one-time GTT and ITT. The authors should show baseline GTT and ITT in the mouse models (before HFD). There is no liver weight; H&E and liver fibrosis results.

Thank you for raising this important point. Like humans, until we observe the early signs of diabetes in post prandial glucose tests or urine tests using dipsticks, we do not perform the oral glucose tolerance test (for human or rats) or GTT and ITT for mouse. Under baseline conditions on a normal chow diet, miR-181c/d KO mice do not show any sign of hyperglycemia by regular urine test using dipstick or by random blood glucose test 1. We performed blood glucose measurement during the mid-light cycle in miR-181c KO mice (6-8 weeks of age) before switching their diet to the HFD. We did not observe any signs of hyperglycemia in miR-181c/d KO mice at this stage. We have now included the data in the results (page 10, second paragraph). Additionally, As shown in Figs. 4B and 4Cs, miR-181c KO mice show no obese phenotype and no changes in plasma triglyceraldehyde with WT mice under aged-matched normal chow diet. Therefore, we anticipate there will be no changes in GTT and ITT at the normal chow condition.

In line 239: “Under baseline conditions when all mice were 6-8 weeks old and on a normal chow diet, light cycle postprandial blood glucose levels did not differ between c/d KO (149.0±4.4 mg/dl) and WT mice (147.4±7.4 mg/dl).”. 

To focus on the role of miR-181c in the liver, we have measured liver weights and performed histology 6 weeks post AAV8-Scr and AAV8-miR-181c injection. As shown in Fig. 10A, there were no differences in the dry liver weights between the two groups. Consistent with Oil-Red-O staining, both the H&E and Mason Trichrome staining show significant lipid droplet accumulation in hepatocytes of animals fed 26 weeks of the HFD (Fig. 10B); but AAV8-miR-181c treatment can normalize lipid accumulation in hepatocytes. Apart from these changes, there were no significant differences in the H&E staining between the AAV8-Scr and AAV8-miR-181c groups. Mason Trichrome staining also shows no significant liver fibrosis (Fig. 10B). 

Minor issue

1. The oil-red staining images have poor quality. No scale bar, Figure 4A is not white balanced.

We apologize for the oversight. We now have included the magnification scale in all pictures. Additionally, we have re-taken all pictures after white balancing. Please see Figs. 4A and 10B. 

Reviewer #2: 

The authors have explored the role of miR-181c in a DIO mouse model using a miR-181c -/- mouse and rescue with a AAV-8 delivering miR-181c.

I do think the observations are of merit and warrant publication but I do have I number of reservations/comments:

Thank you. 

- Be consistent in 181c nomenclature i.e. use 181c throughout

We apologize for the oversight. We now have corrected this I n the manuscript.

- The main text describing Figure 8, I personally found confusing in regards to glucose tolerance test and would just state no blood glucose or insulin levels were measured

We apologize for the confusion. We did measure blood glucose and insulin levels after 10 weeks of the HFD. The first time-point (0 min) of Figs. 8B and 8C represent the fasting blood glucose level and insulin level, respectively. We now have clarified in the text, please see page 14, 2nd paragraph.

 The rational for using the glucose tolerance test (IPGTT) is to check pancreatic-beta-cell function in the AAV8-miR-181c injected group. Because of high adiposity, isolating the pancreas from 10-week HFD-fed mice is difficult. This is why we did not check miR-181c expression in the pancreas of AAV8-miR-181c injected mice (Fig. 7). However, despite using the AAV8 vector, it is possible to overexpress miR-181c in the pancreatic beta-cells. To confirm miR-181c delivery in a liver-specific manner, we have performed IPGTT.

- The authors measure leptin but no rationale as to why measured

The goal of the current study is to establish the role of miR-181c in diet-induced obesity. Plasma leptin level is considered one of the key biomarkers for obesity and metabolic diseases 5,6 which is why we have also measured leptin levels after 26 weeks of the HFD. In order to conclude the protective effects of miR-181c delivery, we have shown that delivery of miR-181c can lower the plasma leptin level compared to the AAV8-Scr injected group (Fig. 10E). We now have included this rationale in the results section (page 11, first paragraph) and again in the discussion section (first paragraph). 

- Why did the authors choose a retro-orbital route for AAV delivery, especially if aiming for liver expression, admittedly this was achieved, but tail vein injection is often used to optimize delivery to the liver.

We use both retro-orbital and tail-vein injections on a regular basis. In our hand, the efficiency of drug delivery into a mouse is better with the retro-orbital route than tail-vein. Therefore, we chose retro-orbital. It has been shown by multiple groups that retro-orbital and tail-vein injection can both be equally effective for delivery into the liver 7,8. 

- Could the authors explain why the numbers of mice in each group or observations, varied so much.

Thank you for this important question. This is a very long protocol – 26 weeks of the HF diet, and we used retro-orbital injection during this time period. During this long-term study, we encountered numerous animal health issues from self-inflected wounds, aggressive behavior with each other or spontaneous dermatitis. In line with the regulations issued by the animal research ethics committee, animals with injuries were euthanized in order to limit suffering. Therefore, even though we start our experiments with equal number of mice (8-10 mice/group), by the end of 26 weeks, we have lost 1 or 2 animals. 

- Figure 4+10 (when any histology) need to include magnification

Sorry for the oversight. We have now included the magnification scale for Figure 4A and 10B.

- Figure 5B +5C can you explain why in whole heart tissue you have low levels of 181c but then can find it in the mito fraction. Whole tissue still has the mito fraction within it, so would it not express 181c too?

Thank you for raising this important point here. The reviewer is absolutely correct that mitochondrial miR-181c expression should be observed in the total heart tissue. However, in Fig. 5B we have compared miR-181c expression in heart and liver tissues. Our data suggest that liver tissue expresses a very high level of miR-181c compared to the heart, and thus, the levels are much higher in Fig. 5B compared to Fig. 5C. Additionally, miR-181c exclusively localizes to the mitochondrial fraction in the heart 9. In Fig. 5C, when we used 1 ng of total RNA from the mitochondrial pellet, miR-181c copy number is significantly higher than 1 ng of total RNA from heart tissue. The total heart tissue derived RNA will have nuclear, cytoplasmic and other cellular organelle fractions apart from mitochondrial RNA. 

- For first use of gene name in the manuscript define, as definition of gene names occurs later in the manuscript or in figure legends

We apologize for the oversight. We have now included the full gene name when they are initially used in the manuscript.

- The evidence for IDH1 is mainly through association based on experimental observations and the literature. It would have been good if the authors had demonstrated that 181c is directly acting on the IDH1 3’UTR in a liver cell to provide more supporting evidence for the authors conclusion.

We agree with the reviewer. However, the dual-luciferase assay to demonstrate the direct association of the miR-181 “seed” sequence and the 3’-UTR of IDH1 mRNA in hepatocytes has already been performed 3. The putative site (UGAAUGU) for miR-181a and miR-181s is identical - that is the “seed” sequence for the miR-181 family 1. Chu et al., 3 has demonstrated that the miR-181 “seed” sequence can directly bind to the 3’-UTR of IDH1 mRNA. Even though the conclusion of the Chu et al., study3 is different than our current study, the dual-luciferase assay is identical to what the reviewer has suggested we do.

- Discussion, I would present some ideas about why 181c has a cytosolic v. mitochondrial location in the liver in contrast to the heart. Are any gender differences anticipated/known, as authors have used a male mouse model. I would be more reserved on the use of 181c as a protective agent for obesity as in the context of the liver it can act as an Oncomir too.

Thank you for the suggestion. We have now discussed this important issue on pages 16-17 within the discussion section.

Additionally, thank you for raising the important question. We now have performed qPCR for miR-181c expression in both age-matched male and female mouse livers. Our results showed that miR-181c expression in female mouse livers is significantly higher than the age-matched male group. 

While it is possible that miR-181c can act as an OncomiR, several studies have shown that miR-181c overexpression can attenuate tumor growth and metastasis for hepatocellular carcinoma (HCC) 4,10. 

References

1. Das S, Kohr M, Dunkerly-Eyring B, et al. Divergent Effects of miR-181 Family Members on Myocardial Function Through Protective Cytosolic and Detrimental Mitochondrial microRNA Targets. J Am Heart Assoc. 2017;6(3).

2. Banavath HN, Roman B, Mackowski N, et al. miR-181c Activates Mitochondrial Calcium Uptake by Regulating MICU1 in the Heart. J Am Heart Assoc. 2019;8(24):e012919.

3. Chu B, Wu T, Miao L, Mei Y, Wu M. MiR-181a regulates lipid metabolism via IDH1. Sci Rep. 2015;5:8801.

4. Mukherjee A, Shrivastava S, Bhanja Chowdhury J, Ray R, Ray RB. Transcriptional suppression of miR-181c by hepatitis C virus enhances homeobox A1 expression. J Virol. 2014;88(14):7929-7940.

5. Ghadge AA, Khaire AA. Leptin as a predictive marker for metabolic syndrome. Cytokine. 2019;121:154735.

6. Boutari C, Mantzoros CS. Adiponectin and leptin in the diagnosis and therapy of NAFLD. Metabolism. 2020;103:154028.

7. Wang F, Nojima M, Inoue Y, Ohtomo K, Kiryu S. Assessment of MRI Contrast Agent Kinetics via Retro-Orbital Injection in Mice: Comparison with Tail Vein Injection. PLoS One. 2015;10(6):e0129326.

8. Price JE, Barth RF, Johnson CW, Staubus AE. Injection of cells and monoclonal antibodies into mice: comparison of tail vein and retroorbital routes. Proc Soc Exp Biol Med. 1984;177(2):347-353.

9. Das S, Ferlito M, Kent OA, et al. Nuclear miRNA regulates the mitochondrial genome in the heart. Circ Res. 2012;110(12):1596-1603.

10. Ai J, Gong C, Wu J, et al. MicroRNA181c suppresses growth and metastasis of hepatocellular carcinoma by modulating NCAPG. Cancer Manag Res. 2019;11:3455-3467.

---

## [Decision Letter · Decision Letter 1]

11 Nov 2021

Role of miR-181c in Diet-Induced Obesity Through Regulation of Lipid Synthesis in Liver

PONE-D-21-26341R1

Dear Dr. Samarjit Das,

We’re pleased to inform you that your manuscript has been judged scientifically suitable for publication and will be formally accepted for publication once it meets all outstanding technical requirements.

Kind regards,

Aijun Qiao, Ph.D.

Academic Editor

PLOS ONE

Additional Editor Comments (optional):

Reviewers' comments:

Reviewer's Responses to Questions

**Comments to the Author**

1. If the authors have adequately addressed your comments raised in a previous round of review and you feel that this manuscript is now acceptable for publication, you may indicate that here to bypass the “Comments to the Author” section, enter your conflict of interest statement in the “Confidential to Editor” section, and submit your "Accept" recommendation.

Reviewer #1: All comments have been addressed

Reviewer #2: All comments have been addressed

2. Is the manuscript technically sound, and do the data support the conclusions?

Reviewer #1: Yes

Reviewer #2: Yes

3. Has the statistical analysis been performed appropriately and rigorously? 

Reviewer #1: Yes

Reviewer #2: Yes

4. Have the authors made all data underlying the findings in their manuscript fully available?

Reviewer #1: Yes

Reviewer #2: Yes

5. Is the manuscript presented in an intelligible fashion and written in standard English?

Reviewer #1: Yes

Reviewer #2: Yes

6. Review Comments to the Author

Reviewer #1: (No Response)

Reviewer #2: The authors have addressed all my major concerns. Though the author's discussion on the organ specific differences of 181c ie heart and liver, would of benefit by being expanded on eg the metabolic function of the liver v the heart primary function of producing sufficient ATP for cardiomyocyte contractility.

7. PLOS authors have the option to publish the peer review history of their article (what does this mean?). If published, this will include your full peer review and any attached files.

Reviewer #1: No

Reviewer #2: No

---

## [Editor Report · Acceptance letter]

24 Nov 2021

PONE-D-21-26341R1 

Role of miR-181c in Diet-induced obesity through regulation of lipid synthesis in liver 

Dear Dr. Das:

I'm pleased to inform you that your manuscript has been deemed suitable for publication in PLOS ONE. Congratulations! Your manuscript is now with our production department. 

Kind regards, 

on behalf of

Dr. Aijun Qiao 

Academic Editor

PLOS ONE